

# Cell-free Synthesis of Proteins with Selectively [13]C-Labelled Methyl Groups from Inexpensive Precursors

Damian Van Raad[1], Gottfried Otting[1,2] and Thomas Huber[1]

[1]Australian National University, Research School of Chemistry, Canberra, ACT 2601, Australia
[2]ARC Centre of Excellence for Innovations in Peptide & Protein Science

*Correspondence to*: Gottfried Otting (Gottfried.otting@anu.edu.au) and Thomas Huber (t.huber@anu.edu.au)

**Abstract.** The novel eCell system maintains the activity of the entire repertoire of metabolic *E. coli* enzymes in cell-free protein synthesis. We show that this can be harnessed to produce proteins with selectively [13]C-labelled amino acids from inexpensive [13]C-labelled precursors. The system is demonstrated with selective [13]C-labelling of methyl groups in the proteins ubiquitin and peptidyl-prolyl *cis–trans* isomerase B. Starting from 3-[13]C-pyruvate, [13]C-HSQC cross-peaks are
obtained devoid of one-bond [13]C-[13]C scalar couplings. Starting from 2-[13]C-methyl-acetolactate, single methyl groups of valine and leucine are labelled. Labelling efficiencies are 70% or higher, and the method allows to produce perdeuterated proteins with protonated methyl groups in residue-selective manner. The system uses the isotope-labelled precursors sparingly and is more readily scalable than conventional cell-free systems.

## 1 Introduction

The NMR resonance assignments of high-molecular weight proteins critically depends on the availability of samples enriched with stable isotopes (Tugarinov et al., 2006). Conventional strategies based on uniformly [13]C-enriched proteins usually employ [U-[13]C]-glucose as the (de facto) only carbon source in minimal media (Filipp et al., 2009). The [13]C-enrichment of proteins enables the sensitive recording of heteronuclear correlation spectra such as 2D [13]C-HSQC spectra, which are particularly

sensitive for methyl groups. Methyl groups play a privileged role in the NMR analysis of large protein systems in solution, as their signals can be observed for macromolecular complexes as large as 1 MDa (Boswell and Latham, 2018). Methyl-bearing amino acids are abundant not only in the hydrophobic core of globular proteins but also in hydrophobic ligand binding pockets (Otten et al., 2010). Methyl groups thus serve as useful probes for the analysis of protein structure, dynamics and function (Schütz and Sprangers, 2020).


Among the amino acids with methyl groups, the spectral regions of the methyl groups of isoleucine, leucine and valine (ILV) overlap in a [13]C-HSQC spectrum (Rasia et al., 2012). This poses a problem for large proteins, which not only contain many methyl groups but also feature broad NMR signals (Lange et al., 2012). Furthermore, uniformly [13]C-labelled proteins feature [13]C-[13]C couplings, in particular one-bond [13]C-[13]C couplings $^1J_{CC}$, which lead to broad multiplets in the [13]C-dimension of [13]C-

HSQC spectra. Several strategies have been devised to resolve the methyl cross-peaks of ILV residues. (i) Protein samples can



be produced from amino acid mixtures containing only a single amino acid with isotope enrichment (Schubert et al., 2006; Schörghuber et al., 2018). Notably, however, suitably labelled amino acids may be available commercially but can be expensive (Takeda et al., 2010). Some of the most affordable versions are uniformly enriched with $^{13}$C, which retains the problem of $^{13}$C-$^{13}$C couplings. Alternatively, isotope-labelled precursors can be supplied, but this usually does not solve the

problem of $^{13}$C-$^{13}$C couplings (Hajduk et al., 2000), and the approach calls for the use of auxotrophic strains to avoid isotope scrambling (Whittaker, 2007; Lazarova et al., 2018). (ii) $^{13}$C-$^1$H correlation spectra can be recorded with homonuclear $^{13}$C-decoupling in the $^{13}$C-dimension, either by recording as a constant-time experiment (Vuister and Bax, 1992), which sacrifices sensitivity, or by band-selective decoupling (Behera et al., 2020), which cannot decouple the $^{13}$C multiplet of leucine methyls as the $^{13}$C chemical shifts of their coupling partners are too close. (iii) Stereospecific-selective labelling of single methyl groups

of each valine and leucine side chain can be achieved by providing 2-$^{13}$C-methylacetolactate in the growth medium (Gans et al., 2010). This method avoids $^1J_{CC}$ couplings and has been successfully applied to protein complexes up to 1 MDa (Sprangers and Kay, 2007), but relies on uncompromised biosynthesis pathways for leucine and valine and thus requires *in vivo* protein production and relatively large quantities of the expensive precursor. The cost issue is magnified for large protein complexes, which are not amenable to multi-dimensional NMR experiments, and where the assignment of the methyl cross-peaks therefore

is achieved by site-directed mutagenesis. A case in point is the 468-kDa multimeric aminopeptidase PhTET2, where the assignment of the alanine $C^\beta H_3$ and isoleucine $C^\delta H_3$ groups alone consumed 3.2 L of media with expensive $^{13}$C-labelled precursors (Amero et al., 2011).

The optimal labelling scheme would be selective by amino acid type, deliver a labelling pattern where the methyl groups are

enriched with $^{13}$C, and their directly bonded carbon is $^{12}$C (Kasinath et al., 2013). Furthermore, it should be amenable to cell-free protein synthesis (CFPS), which uses isotope labelled compounds sparingly (Torizawa et al., 2004). Unfortunately, the biosynthesis of $^{13}$C-labelled amino acids is compromised in *in vitro* protein expression systems (Linser et al., 2014), although a limited degree of metabolism can be restored by re-introducing certain cofactors (Jewett et al., 2008). For example, metabolites from glycolysis can be used for energy generation in CFPS, if cofactors such as NAD$^+$ and CoA are provided (Kim

and Swartz, 2001). Energy generation systems have also been based on phosphoenol pyruvate (PEP), as well as pyruvate, glucose and maltodextrin (Caschera and Noireaux, 2015). In our hands, these systems proved to be more difficult to establish presumably because of their dependence on the activity of multiple enzymes from the glycolytic pathway. The problem of maintaining the activity of enzymes required for energy regeneration in CFPS has been solved, however, by the recently established eCell system (Van Raad and Huber, 2021).


eCells are bacterial cells coated with polymers, where the cell wall has been lysed (Van Raad and Huber, 2021). The resulting cells can no longer replicate, but they still contain all biomacromolecules required for protein synthesis, while their porous polymer coat gives low-molecular weight compounds free access to the cytosol. eCells thus are ideal vehicles for CFPS. We hypothesized that eCells also preserve the activity of the enzymes involved in the biosynthesis of amino acids and therefore



allow the production of methyl-labelled amino acids from inexpensive precursors such as 3-$^{13}$C-pyruvate or $^{13}$C-glucose. The present work demonstrates that this is indeed the case.

Different precursors can be used for selective $^{13}$C-enrichment of ILV methyl groups. Well-established isotopologue precursors used in *in vivo* protein expression include α-ketobutyrate and α-ketoisovalerate (Goto et al., 1999), which is a key precursor
for both leucine and valine synthesis (Lundström et al., 2007).  In general, precursors close to the final stages of biosynthesis of methyl bearing amino acids allow the labelling of proteins with high selectivity (Schörghuber et al., 2018). The last precursor for selective stereospecific methyl labelling of leucine and valine is 2-methyl-4-acetolactate (Figure 1B), whereas early precursors in the biosynthetic pathways of amino acids, such as pyruvate, cannot label methyl groups in a stereospecific manner. In the following we demonstrate the excellent utility of eCells to produce proteins with selectively $^{13}$C-labelled methyl
groups from different inexpensive precursors, including pyruvate, 2-methyl-4-acetolactate and glucose.









**Figure 1.** Biosynthetic pathways of leucine and valine from isotope-labelled precursors. [13]C-labelled methyl groups are identified by green
balls, and methyl groups at natural isotopic abundance are highlighted by orange balls. (A) Biosynthetic pathway starting from 3-[13]C-labelled
pyruvate. (B) Stereoselective biosynthetic pathway starting from (S)-2-acetolactate. Abbreviations used: KARI, ketol-acid reductoisomerase;
DAD, dihydroxy-acid dehydratase; IMS, 2-isopropylmalate synthase; LEU1, 3-isopropylmalate dehydratase; IMDH, 3-isopropylmalate
dehydrogenase; BCAT, branched-chain aminotransferase.

## 2 Materials and methods

### 2.1 Materials

The polyelectrolytes low-molecular weight chitosan (50,000 – 190,000 Da) and sodium alginate were purchased from Merck.
The ethyl ester of 2-[13]C-methyl-4-[2]H$_3$-acetolactate (ethyl-2-hydroxy-2-[13]C-methyl-3-oxobutanoate) was purchased from
Cambridge Isotope Laboratories (CIL; USA). Perdeuterated amino acids were from CIL and Martek Isotopes (USA). 3-[13]C-
pyruvate was from Sigma-Aldrich.

### 2.2 Plasmids

A plasmid was constructed with the pCloDF13 origin of replication, the gene of the *E. coli* peptidyl–prolyl *cis–trans* isomerase
PpiB with C-terminal His$_6$-tag under control of the T7 promoter and a spectinomycin resistance gene, generating the plasmid
pCDF PpiB CTH. For ubiquitin expression a plasmid was constructed with pCloDF13 origin of replication, the spectinomycin
resistance gene and the gene of ubiquitin under control of the T7 promoter (plasmid pCDF Ubi CTH). A *lac* operator was
inserted in front of the T7 promoter to reduce background protein expression prior to induction which reduces the [13]C labelling
efficiency.

### 2.3 Production of eCells

*E. coli* Xjb cells were transformed with either pCDF Ubi CTH or pCDF PpiB CTH and grown in LB medium at 37 °C in
baffled flasks with shaking at 180 rpm. Endolysin production was induced at the time of inoculation with a final concentration
of 3 mM arabinose. Cells were grown to OD$_{600}$ = 0.6, harvested by centrifugation and washed three times with PBS-E buffer
(phosphate-buffered saline with 1 mM EDTA, pH 7.4). For coating with chitosan, the cells were resuspended in 0.25 mg/mL
chitosan in PBS-E with vigorous shaking for 20 minutes. The cell pellet was washed with PBS-E pH 6.0 three times to remove
excess chitosan and then resuspended in 0.25 mg/mL alginate PBS-E solution and subjected to vigorous shaking for 20
minutes. The cells were then washed 3 times with PBS-E pH 6.0, resuspended in PBS-E pH 7.4 and stored at -80 °C in PBS-
E buffer.

### 2.4 Production of deuterated eCells

5 g sodium pyruvate was dissolved in 50 mL $D_2O$ and the pH adjusted with 0.1 mM KOD to pH 11. The solution was stirred overnight at 95 $^oC$ to exchange the protons of pyruvate for deuterium. 500 mL M9 minimal media was prepared in $D_2O$ with 22 mM $KH_2PO_4$, 42 mM $Na_2HPO_4$, 8.6 mM NaCl, 18.6 mM $NH_4Cl$, 500 μL 1 mg/mL thiamine (vitamin B6), 0.1 mM $CaCl_2$, 250 μL 1000x metal mixture (50 mM $FeCl_3$, 10 mM $MnCl_2$, 10 mM $ZnSO_4$, 2 mM $CoCl_2$, 2 mM $CuCl_2$ and 2 mM $NiCl_2$), 5

mM $MgSO_4$, 3 mM arabinose and 25 mg/mL spectinomycin. The H-D exchange in pyruvate was confirmed by NMR. The deuterated pyruvate was added to the dry mixture of buffer salts and the final pyruvate-M9 medium made up to 500 mL, adjusted to pH 7.2 and filter sterilised prior to inoculation.

XjB(DE3)* cells that had been transformed with pCDF PpiB CTH were trained for production of perdeuterated proteins in a

protocol adapted from that reported by Li and Byrd (2022). 15 mL of an overnight starter culture of pCDF PpiB CTH was diluted with 15 mL of deuterated pyruvate-M9 medium and incubated at 37 $^OC$ with shaking at 180 rpm. When the $OD_{600}$ reached 1.0, the cells were again diluted with 30 mL of deuterated pyruvate-M9 medium and incubated a second time. Upon reaching $OD_{600}$ = 1.0, the 60 mL culture was spun down, the cells transferred to a 50 mL culture and growth continued overnight at 37 $^OC$ with shaking at 180 rpm. The 50 mL culture was added to 400 mL of deuterated pyruvate-M9 medium and

left to grow until $OD_{600}$ = 0.75 was reached, after which the cells were encapsulated as described in Section 2.3.

### 2.5 CFPS systems

The protocol for pyruvate-based CFPS was adapted from the phosphate recycling system by Jewett and Swartz (2004). The CFPS buffer contained 0.9 mM UTP and CTP, 50 mM HEPES, 1.5 mM GTP, 1.5 mM ATP, 0.68 μM folinic acid, 0.64 mM

cAMP, 1.7 mM DTT, 3.5 mM of each amino acid (apart from the amino acid(s) to be synthesized by the eCells for isotope enrichment), 60 mM K-Glu, 8 mM Mg-Glu, 2% v/v PEG-8000, 4 mM sodium oxalate, 0.25 mM CoA and 0.33 mM $NAD^+$. Roche cOmplete$^{TM}$ Mini protease inhibitor cocktail was added to the CFPS buffer. Of the volume following dissolution of one tablet in 10 mL water, 10% were added to the CFPS reaction. The reaction was conducted with 33 mM pyruvate.

The protocol for glucose-based CFPS was likewise adapted from the previously published phosphate recycling system (Jewett and Swartz, 2004). The glucose CFPS buffer contained the same components as the pyruvate-based CFPS protocol, but with 10 mM sodium phosphate dibasic pH 7.5 and without sodium oxalate and pyruvate. The reaction was conducted with 30 mM glucose.

The CFPS system using creatine phosphate and creatine kinase as energy source contained the same components as the pyruvate-based CFPS protocol, but without sodium oxalate, pyruvate, CoA or $NAD^+$ and adding 250 μg/mL creatine kinase, 80 mM creatine phosphate and 6 mM Mg-Glu instead of 8 mM Mg-Glu.



**MAGNETIC RESONANCE**
Discussions

Following addition of the isotopically enriched precursor, the CFPS buffers for each of these reactions were adjusted to pH

7.5. Frozen aliquots of encapsulated cells were thawed and the pellet immersed in CFPS buffer. CFPS for each experiment was conducted at 37 °C with shaking at 180 rpm.

### 2.6 Acetolactate labelling

2-$^{13}$C-methyl-4-$^2$H$_3$-acetolactate as the source for prochiral methyl groups was set free from the ethyl ester by incubating in

H$_2$O with 0.1 M NaOH (NaOD for deuteration experiment) (pH 13) at 37 °C for 30 minutes. The compound was tested both in pyruvate-based CFPS and in CFPS with the creatine-phosphate/creatine kinase system. The CFPS reaction was conducted in 15 mL buffer with 0.1 mM NADP$^+$, 3.5 mM 2-$^{13}$C-methyl-4-$^2$H$_3$-acetolactate and 0.2 mM penoxsulam to inhibit the acetolactate synthase (ALS) enzyme. Ubiquitin was produced from 300 mg eCells and purified using His-Gravitrap columns (GE Healthcare, USA).


For perdeuterated CFPS, all buffer stocks were dissolved in D$_2$O, and the pH was adjusted with KOD to pH 7.2. The creatine phosphate based CFPS reaction was conducted in 20 mL D$_2$O buffer with 5 mM 2-$^{13}$C-methyl-4-$^2$H$_3$-acetolactate, 0.1 mM NADP$^+$, 1 mM of all amino acids in perdeuterated form excluding valine and 0.2 mM penoxsulam to inhibit the acetolactate synthase (ALS) enzyme. PpiB was produced from 800 mg eCells and purified using His-Gravitrap columns (GE Healthcare,

USA).

### 2.7 Labelling with 3-$^{13}$C-pyruvate or 1-$^{13}$C glucose

Dry 3-$^{13}$C-pyruvate was added to 15 mL CFPS buffer at 33 mM final concentration. Leucine, valine and isoleucine were omitted from the amino acid mixture to allow for $^{13}$C-labelling of their methyl groups. Ubiquitin and PpiB were expressed

using 300 mg eCells and purified using His-Gravitrap columns. To illustrate the scalability of the reaction, ubiquitin samples were also produced with specific labelling of alanine and valine in 5 mL CFPS buffer using 300 mg eCells with the amino acid of interest omitted from the amino acid mixture. PpiB with $^{13}$C-labelled valine was produced in 20 mL pyruvate-based CFPS buffer with 1 g eCells with valine omitted from the amino acid mixture.

To test the performance of 1-$^{13}$C-glucose as $^{13}$C source, dry 1-$^{13}$C glucose was added to 5 mL glucose-based CFPS buffer at 30 mM final concentration. Leucine and valine were omitted from the amino acid mixture to allow for labelling of their methyl groups. To assess the potential of glutamate in the buffer in diluting the $^{13}$C-label, reactions were conducted with a buffer containing 60 mM K-Glu/8 mM Mg-Glu or 100 mM adipic acid/8 mM MgCl$_2$.

### 2.8 NMR spectroscopy and isotope labelling yields

All NMR spectra were recorded at 25 °C using 5 mm NMR tubes and a Bruker 800 or 600 MHz NMR spectrometer equipped with TCI cryoprobes.



The isotope labelling efficiency of leucine residues in ubiquitin was assessed by integrating the $^1$H-NMR signals of the $\delta_2$-
methyl group of Leu50 and its $^{13}$C satellites, which are resolved in the 1D NMR spectrum. For samples without isotope-labelled leucine, the $^{13}$C-HSQC cross-peak intensities of the labelled residues were compared with those of an internal standard of 0.1 mM 3-$^{13}$C-pyruvate.

## 3 Results

### 3.1 Ubiquitin with $^{13}$C-labelled methyl groups in alanine, leucine and valine made from 3-$^{13}$C-pyruvate

The biosynthetic methyl labelling strategies were validated using ubiquitin as a model protein. The $^{13}$C-label was provided by 3-$^{13}$C-pyruvate, which served both as carbon source for amino acid synthesis and energy source for protein production. Omission of leucine and valine from the reaction mixture allows for detection of $^{13}$C-labelled valine and leucine produced from pyruvate during the cell-free reaction. As singly $^{13}$C-labelled pyruvate contains no neighbouring $^{13}$C atoms, the methyl

groups of leucines and valines are expected to not show any $^1J_{CC}$ coupling. This expectation was borne out in the experiment, where the cross-peaks revealed no splittings in the $^{13}$C-dimension (Figure 2). Therefore, this labelling scheme delivers better spectral resolution than uniform $^{13}$C-labelling schemes, where multiplet splittings due to $^1J_{CC}$ couplings can be avoided only by specific pulse sequences that compromise sensitivity (Vuister and Bax, 1992; Behera et al., 2022). As the biosynthetic pathways remained intact, the $^{13}$C-label was subject to incorporation into a range of amino acids and thus prone to some isotope

scrambling. For example, isotopic enrichment was also detected for alanine (due to direct conversion of pyruvate to alanine by alanine-transaminase) and the $\gamma_2$-methyl group of isoleucine. Importantly, however, the labelling efficiency of the isopropyl groups of leucine and valine was high (about 70%).





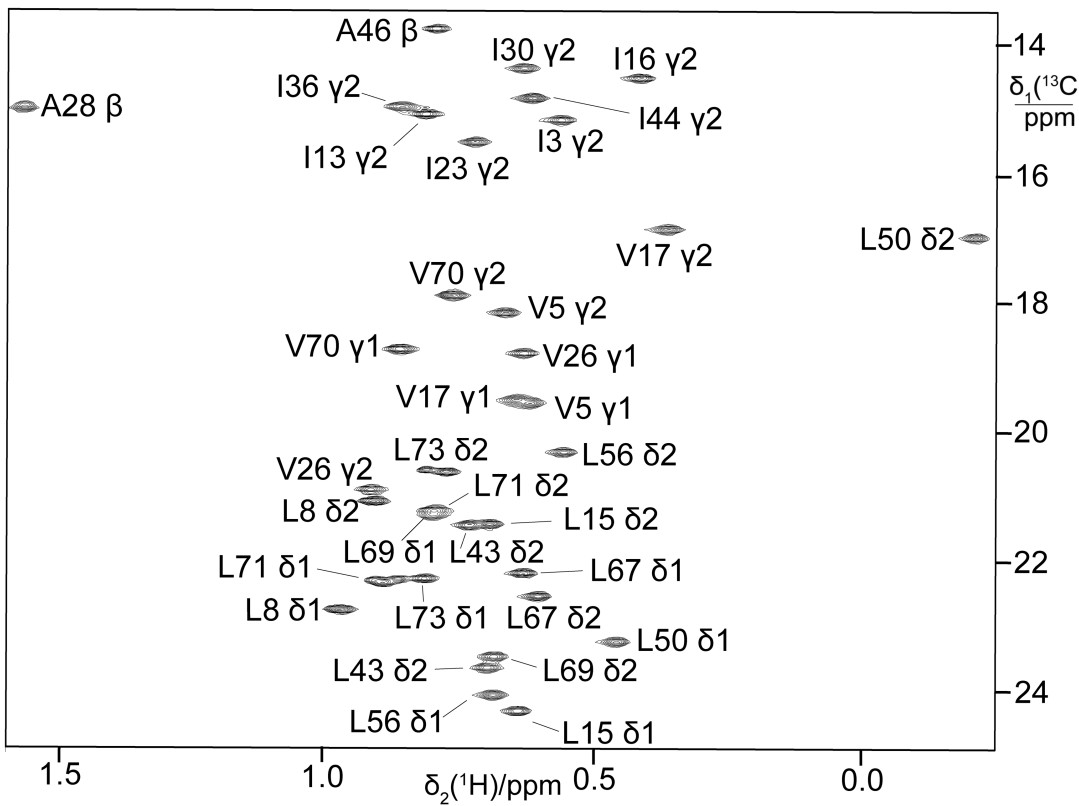

**Figure 2.** $^{13}$C-HSQC spectrum of ubiquitin produced from 3-$^{13}$C-pyruvate by eCell CFPS, resulting in uniform $^{13}$C-labelling of both isopropyl methyl groups of leucine and valine. The protein yield was 0.7 mg from 10 mL eCell CFPS reaction, and the level of isotope labelling was 70%.

The absence of $^{13}$C-enrichment of the δ-methyl group of isoleucine is a signature of the biosynthetic pathway, where one pyruvate molecule is linked with unlabelled acetyl-CoA to form α-ketobutyrate as the precursor of isoleucine, channelling the $^{13}$C label into the γ$_2$-methyl rather than the δ-methyl group.

As the cell-free reaction was performed in a buffer containing high concentrations of glutamate, we speculated that the degree of isotope labelling could be increased by substituting glutamic acid for adipic acid, which is not easily converted into amino acids (Jia et al., 2009). Using 100 mM adipic acid/8 mM MgCl$_2$ instead of 60 mM K-Glu/8 mM Mg-Glu, however, did not increase the labelling efficiency and slightly decreased the protein yield (data not shown).



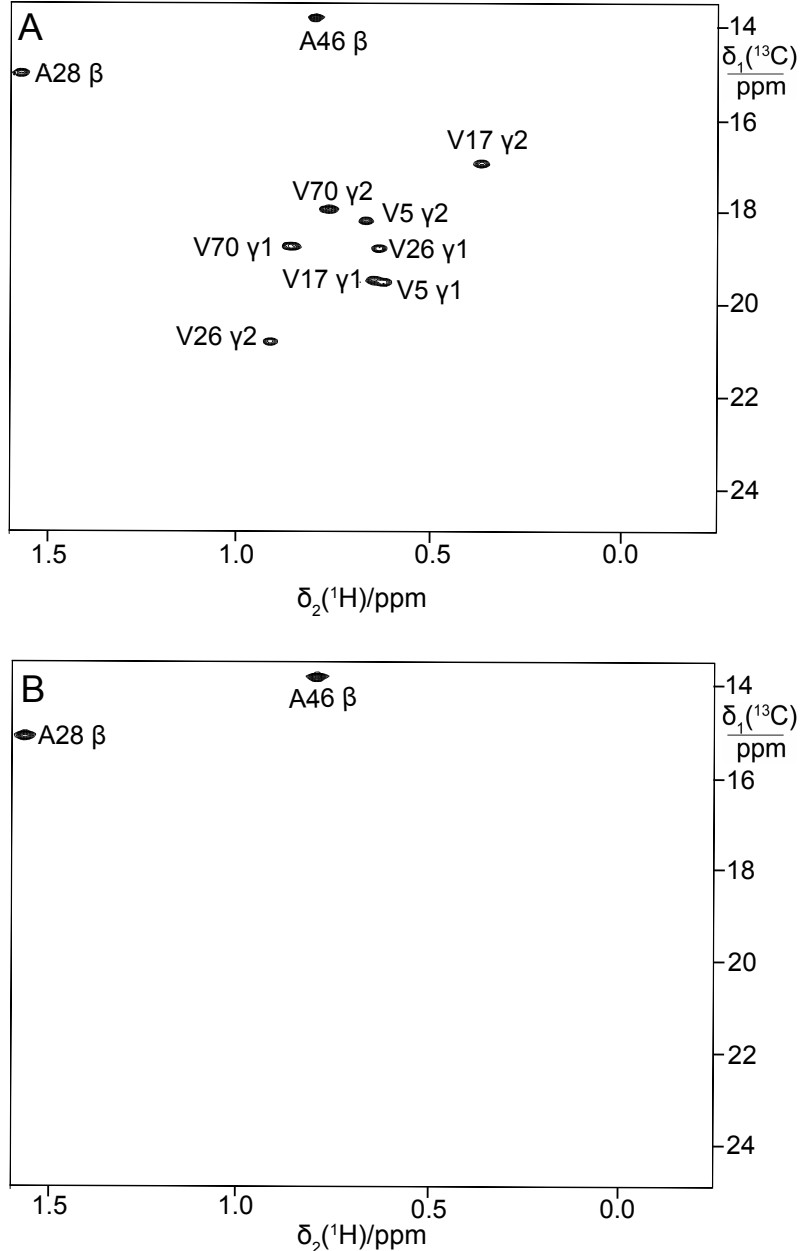

**Figure 3.** $^{13}$C-HSQC spectra of ubiquitin expressed in a 5 mL CFPS reaction using 300 mg eCells with 3-$^{13}$C-pyruvate. (A) Valine was omitted from the amino acid mixture. Protein yield 1.6 mg, $^{13}$C-enrichment of the valine methyl groups >70%. (B) Alanine was omitted from the amino acid mixture. Protein yield 1.35 mg, $^{13}$C-enrichment of the alanine methyl groups >85%.




Starting from 3-$^{13}$C-pyruvate for biosynthesis, the selectivity of isotope labelling was enhanced by 'unlabelling' the amino acids not of interest for labelling, which is achieved simply by adding them to the CFPS reaction at natural isotopic abundance. For example, the $^{13}$C label was apparent only in the valine methyl groups when only valine was omitted from the amino acid mixture (Figure 3A), and only alanine peaks were observed when only alanine was left out (Figure 3B).


**3.2 Ubiquitin with $^{13}$C-labelled methyl groups in leucine and valine made from 2-$^{13}$C-methyl-acetolactate**

2-$^{13}$C-methyl-acetolactate has been shown to allow the *in vivo* production of proteins with stereospecifically labelled isopropyl groups of valine and leucine (Gans et al., 2010). To test the performance of this approach with eCells, a sample of ubiquitin was prepared with the provision of 2-$^{13}$C-methyl-acetolactate and penoxsulam, which is a bactericidal acetolactate synthase

(ALS) inhibitor that blocks the biosynthetic conversion of pyruvate to acetolactate, thus abolishing the synthesis of leucine and valine from pyruvate. Both the unlabelled pyruvate and creatine phosphate ATP regeneration systems were used. Both resulted in stereoselective labelling with similar labelling efficiency, highlighting the absence of any significant isotopic dilution by the addition of pyruvate at natural isotopic abundance. Figure 5 shows that the prochiral *S*-methyl groups of ubiquitin were stereoselectively labelled as expected. Although the ALS inhibitor did not entirely prevent the incorporation of

unlabelled valine and leucine, presumably due to the unlabelled amino acids already present in the eCells prior to protein production, the isotope labelling efficiency nevertheless reached 70%. Importantly, the eCell system enabled production of this selectively $^{13}$C-labelled sample from less than 6 mg methyl-acetolactate precursor, and no $^{13}$C labelling of pro-*R* methyl groups was detectable. The effectiveness of the ALS inhibitor in preventing the production of unlabelled valine and leucine was confirmed by comparison with the isotope labelling efficiency when the CFPS was performed using the widely used ATP-

regeneration system with creatine phosphate and creatine kinase (Kigawa et al., 1999; Apponyi et al., 2008). The same isotope labelling efficiency was obtained and the same protein yield (0.7 mg).



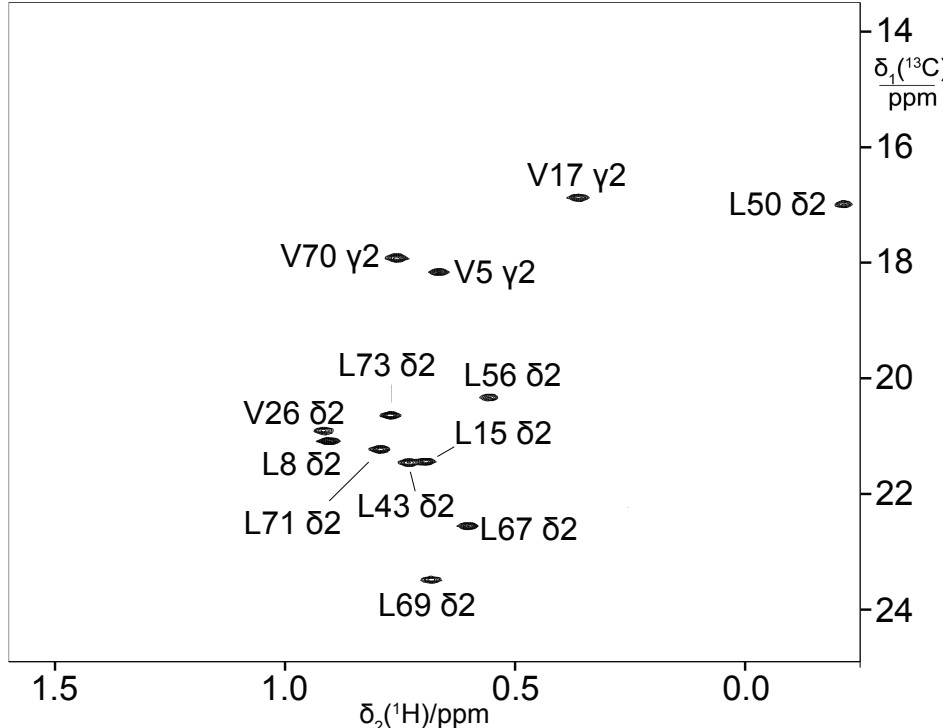

**Figure 4.** $^{13}$C-HSQC spectrum of ubiquitin with labelling of the pro-*S* methyl groups in leucine and valine by using site-specifically $^{13}$C-labelled acetolactate in eCell CFPS. Protein yield 0.7 mg, isotope labelling efficiency 70%.


### 3.3 PpiB with stereospecific $^{13}$C-labelled methyl groups in valine

To illustrate the broad applicability of the eCell approach to produce perdeuterated proteins, it was also applied to the *E. coli* peptidyl-prolyl *cis–trans* isomerase B (PpiB), which is a 19 kDa protein. Figure 5A shows the $^{13}$C-HSQC cross-peaks of PpiB prepared with 3-$^{13}$C-pyruvate while omitting valine. Although the methyl groups of alanine residues are also observed, no two
cross-peaks overlap to the extent that they cannot be recognized as separate cross-peaks.

Figure 5B shows the $^{13}$C-HSQC cross-peaks of perdeuterated PpiB made by eCell CFPS using perdeuterated eCells, pyruvate deuterated by H-D exchange and 2-$^{13}$C-methyl-4-$^{2}$H$_3$-acetolactate. All amino acids were provided in perdeuterated form and valine was omitted. This resulted in stereoselective labelling of the pro-*S* groups of valine residues in PpiB with a high labelling
efficiency (ca. 90%) and adequate yield (1.32 mg). The deuteration level of the protein was high, as shown by a 1D $^{1}$H-NMR spectrum (Figure in SI).



**Figure 5.** Selective $^{13}C$-labelling of the methyl groups of alanine and valine residues in PpiB produced by eCell CFPS. (A) $^{13}C$-HSQC spectrum of PpiB produced from 3-$^{13}C$-pyruvate with valine omitted. Published assignments are shown (BMRB file 11451). The spectrum



also displays the cross-peaks of the $\gamma_1$-methyl groups, but their assignments have not been reported. Protein yield 2.2 mg, isotope labelling efficiency >75%. (B) $^{13}$C-HSQC spectrum of PpiB produced from 2-$^{13}$C-methyl-acetolactate by CFPS with valine omitted in an eCell CFPS reaction in D$_2$O using deuterated eCells. The $^{13}$C-HSQC spectrum illustrates the selective labelling of the pro-*S*-methyl groups of valine in a perdeuterated protein. The protein yield was 1.3 mg, and the $^{13}$C-labelling level was 90%.

**3.4 eCell CFPS for stereospecific assignments by biosynthetically directed fractional $^{13}$C-labelling**


Biosynthetic fractional $^{13}$C-labelling is a well-established approach to obtain stereospecific assignments of isopropyl methyl groups (Senn et al., 1989; Neri et al., 1989). Starting from a mixture of 10% uniformly $^{13}$C-labelled glucose and 90% glucose at natural isotopic abundance, the $^{13}$C-NMR spectrum of pro-*R* methyl groups displays splittings due to $^1J_{CC}$ couplings while the pro-*S* methyl groups do not. The approach is inexpensive as only little isotope-labelled glucose is needed. To explore

whether eCells maintain the required biosynthetic pathway, a sample of ubiquitin was prepared from a mixture of $^{13}$C-labelled and unlabelled glucose. The $^{13}$C-HSQC spectrum showed the multiplet finestructures expected for the pro-*R* and pro-*S* methyl groups (Figure 6).

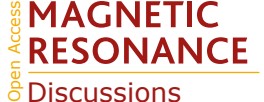

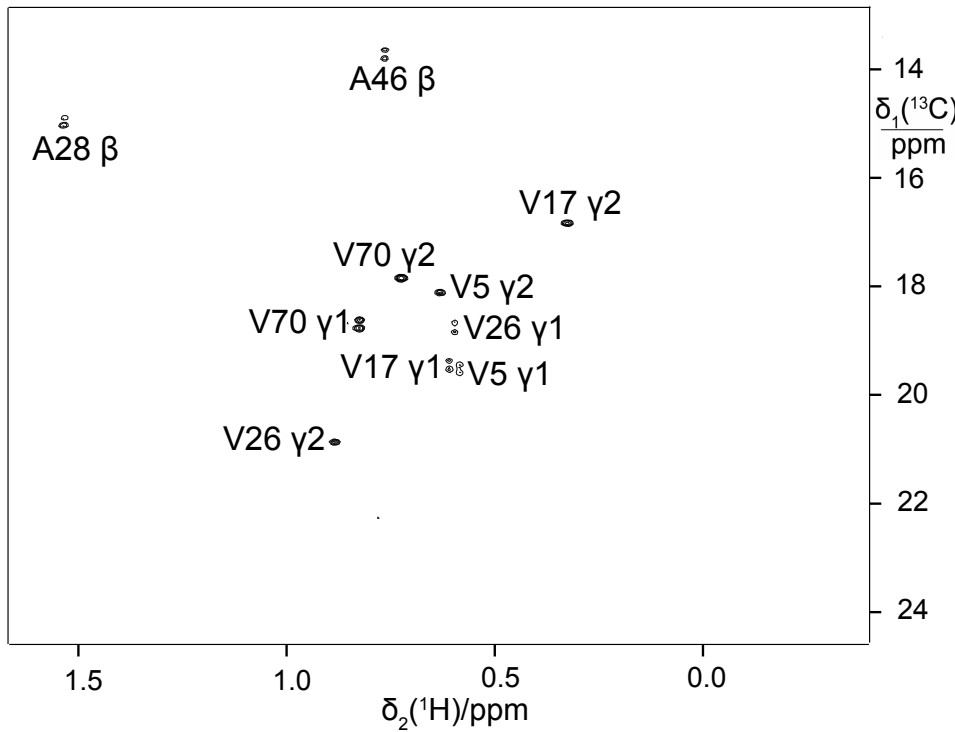

**Figure 6.** $^{13}$C-HSQC spectrum of ubiquitin produced by eCell CFPS from a mixture of 10% uniformly $^{13}$C-labelled glucose and 90% glucose at natural isotopic abundance. Protein yield 3.8 mg, labelling efficiency below 10%.

**4 Discussion**

Biosynthetic pathways naturally established in bacterial cells can be exploited to produce selectively $^{13}$C-labelled proteins in a cell-free reaction that delivers protein in yields sufficient for NMR analysis. In contrast to conventional CFPS reactions, which are based on cell extracts prepared by high-speed centrifugation, the eCells used in the present work much better preserve the activities of the natural complement of biosynthetic enzymes of the parent live *E. coli* cells. In this way, the eCells combine the potential inherent in biosynthetic pathways with the advantages of conventional CFPS, namely the low requirement of amino acids (Torizawa et al., 2004), the compatibility with toxic proteins (Martemyanov et al., 2001) and the facile modification of the solution conditions for protein production. Importantly, eCells can be produced much faster than the cell extracts required for conventional CFPS. Table 1 highlights the low cost for isotope-labelled precursors used in the experiments of the present work.




**Table 1.** Comparison of precursors and their contribution to the cost of eCell CFPS reaction with 300 mg eCells.

| Precursor and energy source | Precursor cost (USD) | Labelling degree | Protein yield (mg/ml) | Cost per reaction (USD) | Position labelled |
|---|---|---|---|---|---|
| 2-$^{13}$C-methyl-4-$^{2}$H$_3$-acetolactate | $1722/g | 90% | 0.7 | $9 | V= $\gamma_2$ <br> L= $\delta_2$ |
| 3-$^{13}$C-pyruvate | $866/g | 70% | 0.8 | $34 | V= $\gamma_2$, $\gamma_1$ <br> L= $\delta_2$, $\delta_1$ <br> I= $\gamma_2$ |
| 10% [U-$^{13}$C]-glucose + 90% unlabelled glucose | $258/g | 10% | 3.8 | $2 | V= $\gamma_2$ |
| 1-$^{13}$C-glucose | $282/g | 44% | 1.4 | $14 | V= $\gamma_2$, $\gamma_1$ <br> L= $\delta_2$, $\delta_1$ |

[1] Prices from Cambridge Isotope Laboratories (https://www.isotope.com) and Omicron Biochemicals, Inc. (https://www.omicronbio.com), accessed 4 Jan 2023.


The present work demonstrates the use of eCells for selective, and also stereospecific, $^{13}$C-labelling of the methyl groups of leucine and valine. $^{13}$C-labelled methyl groups are privileged for NMR investigations of large proteins, as their magnetization relaxes more slowly than the magnetization of any other amino acid moiety (Tugarinov and Kay, 2005). Resolving the spectral overlap between the $^{13}$C-HSQC cross-peaks of different methyl groups, however, is not straightforward, as the $^{13}$C-NMR

spectra of proteins produced from uniformly $^{13}$C-enriched precursors feature multiplet splittings governed by the large $^{1}J_{CC}$ coupling constant. In principle, the splitting could be avoided if the protein were made from suitably isotope-labelled amino acids, but leucine and valine with stereospecific single $^{13}$C-enrichment of only one of the isopropyl methyl groups are prohibitively expensive or demanding to prepare (Linser et al., 2014).

As stereospecific $^{13}$C-labelling achieved by *in vivo* protein expression with 2-$^{13}$C-methyl-4-acetolactate (Gans et al., 2010) has become very popular, this compound has become available commercially. (We found the deuterated isotopologue 2-$^{13}$C-methyl-4-$^{2}$H$_3$-acetolactate to be more readily available than the undeuterated analogue, but the selective $^{13}$C-labelling strategy would be beneficial also without deuteration.) Our results show that this compound can be used in protein production by eCell CFPS, and the resulting $^{13}$C-HSQC spectra clearly identify the pro-*S* methyl groups. Alternatively, eCell CFPS also allows the

stereospecific distinction of the isopropyl methyl groups by the classical method of biosynthetic fractional $^{13}$C-labelling that

uses an inexpensive mixture of uniformly $^{13}$C-labelled glucose with an excess of glucose at natural isotopic abundance (Neri et al., 1989). Although this scheme allows stereospecific assignments at extraordinarily low cost as far as $^{13}$C-labelled glucose is concerned, the level of isotope labelling associated with this scheme is intrinsically low, and we therefore prefer 2-$^{13}$C-methyl-4-acetolactate for stereospecific assignments, which also minimizes cross-peak overlap by avoiding $^1J_{CC}$ multiplet

splittings.

The present study also outlines a protocol for the cell-free synthesis of perdeuterated proteins utilizing eCells generated from deuterated media. The approach is eminently practical and economical. Additional versatility arises from the straightforward way, in which spectral simplification can be obtained by the provision of deuterated amino acids. The high levels of $^{13}$C

incorporation (90%) and deuteration (estimated to be >95%) are comparable with *in vivo* protein preparations and favourable for good sensitivity of NMR experiments of large protein complexes (O'Brien et al., 2018).

Pyruvate plays a central role in bacterial biosynthesis and, as shown in the present work, singly $^{13}$C-labelled pyruvate is suitable as a relatively inexpensive precursor for labelling methyl groups of leucine and valine with high levels of $^{13}$C-enrichment. If,

at the same time, unlabelled leucine or valine is provided in the CFPS reaction to suppress their respective cross-peaks, the cross-peaks of the amino acid omitted can be observed selectively. The increased spectral resolution afforded by this scheme is particularly beneficial for larger proteins. Furthermore, inactivation of transaminases by reduction with NaBH$_4$ (Su et al., 2011) may allow extending this approach to the selective $^{15}$N-labelling of amino acids from $^{15}$N-ammonium salt. These experiments are currently in progress.


In principle, using 1-$^{13}$C-glucose as the carbon source delivers the same selectivity of isotope labelling as 3-$^{13}$C-pyruvate (Lundström et al., 2007) but, as glycolysis breaks the glucose down into 3-$^{13}$C-pyruvate and unlabelled pyruvate, glucose simultaneously labelled in the 1 and 6 position is required to avoid the dilution with unlabelled pyruvate (Loquet et al., 2011). We therefore prefer 3-$^{13}$C-pyruvate.


As pyruvate can be converted to alanine by a single enzyme, it is difficult to suppress the cross-peaks of the C H$_3$ groups of alanine when starting from $^{13}$C-labelled pyruvate. The addition of an excess of unlabelled alanine to the reaction would dilute the labelled pyruvate with unlabelled pyruvate, and inhibition of the alanine aminotransferase by reduction with NaBH$_4$ would also inhibit the transaminase that installs the amino group on leucine and valine by transfer from glutamate. We therefore

propose to identify the alanine cross-peaks with a sample, where the isotope labelling of leucine and valine is suppressed by provision of these amino acids in unlabelled form (Figure 3B).

Starting from pyruvate, we found it difficult to achieve labelling efficiencies much above 70%. We attribute this to an isotope dilution effect due to a pool of unlabelled amino acids present in the eCells. Attempts to dialyse eCells in a large volume of

buffer for an extended period of time reduced the protein yield as the eCells lose activity by gradually leaking biomacromolecules (Van Raad and Huber, 2021). Notably, proteins produced *in vivo* from various $^{13}$C-labelled glucose isotopomers are likewise subject to isotopic dilution, and examples with ~45% labelling efficiency have been reported (Lundström et al., 2009; Loquet et al., 2011; Weininger, 2017).

As proteins slowly leak through the porous polymer coating, the lifetime of eCells is limited to about 8 h, which limits protein yields. It is important to note, however, that *in vivo* protein expression from selectively labelled precursors cannot be conducted for too long either, if isotope scrambling by precursor recycling is to be avoided (Kurauskas et al., 2017).

**5 Conclusions**

In summary, the eCell platform opens new possibilities for the selective $^{13}$C-enrichment of methyl groups in proteins. It combines high levels of isotope enrichment with low cost of isotope-enriched precursors. The protocol for eCell preparation is uncomplicated, and the open nature of eCells provides exquisite control over the chemical environment, so that different isotope labelling is achieved simply by use of different reaction buffers. In contrast to conventional CFPS based on dialysis systems, where protein yields depend on good contact between inside and outside buffers and, therefore, the geometry of the 355 setup, the eCell system can readily be scaled in volume. We anticipate that it will find many more uses beyond those demonstrated in the present work.

**Data availability.** The NMR data are available at https://doi.org/10.5281/zenodo.7662927 (Van Raad et al., 2023).

**Supplement.** The supplement contains the nucleotide sequences of genes used in this work, and high-resolution mass spectrum of the deuterated PpiB sample.

**Author contributions.** DVR initiated the project and performed all biochemical experiments. GO performed all NMR experiments. GO and TH coordinated the project and contributed advice towards experimental design. All three authors 365 contributed to the final manuscript.

**Competing interests.** The Australian National University holds a patent related to the production and use of eCells (PCT/AU2020/050050) and share financial return from the patent with the inventors (TH and DVR). At least one of the (co-)authors is a member of the editorial board of *Magnetic Resonance*.


**Acknowledgements.** Gottfried Otting thanks the Australian Research Council, for a Laureate Fellowship (grant no. FL170100019).



**Financial support.** This research has been supported by the Australian Research Council (grant no. FL170100019,
DP200100348, DP21010088) and the Australian Research Council Centre of Excellence for Innovations in Peptide and Protein
Science (grant no. CE200100012).

**NMR spectra are available on Zenodo:** https://doi.org/10.5281/zenodo.7662927 (Van Raad et al., 2023)

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
