# Peer review of "Cell-free Synthesis of Proteins with Selectively 13C-Labelled Methyl Groups from Inexpensive Precursors"

_Magnetic Resonance, 2023_

## Author Response (AR1)

We thank the referees for the positive comments. In response to the specific points raised, we have prepared a revised version of the manuscript as outlined below, with changes made highlighted in yellow in a highlighted copy of the main text of the revised manuscript.

Referee 2:

1. **Methyl-specific labeling of proteins in vitro using 2-keto acids has been previously reported in the literature (Lazarova et al, ACS Chem. Biol. 2018, 13, 2170-2178), including the standard 2-ketoisovalerate for valine labeling. This should be clearly stated in the introduction and commented in the discussion section.**

   We apologize for quoting this reference in a misleading way. We have rewritten the introduction to correct for this.

2. **The most useful methyl probes in large proteins are ile-d1. Can the proposed approach be used with standard 2-ketobutyrate for specific labeling of ile-d1 methyl groups? The papers will clearly be improved if these data are included.**

   We are confident that it would work but have no experimental proof and therefore cannot give a definite answer at present. Starting from 2-ketobutyrate requires five enzymatic steps to produce isoleucine. As the lead author performing the work has completed their PhD and is no longer in the laboratory, it is difficult for us to include experimental results on isoleucine labelling in the present manuscript.

3. **Methyl-specific labeling is very important for the NMR studies of very large proteins, but it requires the simultaneous perdeuteration of proteins. In vitro, this requires the use of perdeuterated amino acids at a concentration of 1 mM for each amino acid in addition to the methyl-labeled amino acids or precursors. Reconstitution of these mixtures can be expensive and the cost of the 2H amino acids used should be considered in the cost estimates in Table 1 and in the discussion section**

   We have amended the Discussion section to discuss the cost of 2H-labelled amino acids and now provide a detailed breakdown of the amino acid costs in Table S1 of the Supporting Information. Uniformly deuterated amino acids add to the cost of preparing samples but are still affordable as the volume of cell-free protein synthesis (CFPS) reactions is small.

4. **Lines 34-35: "Alternatively, isotope-labeled precursors can be provided, but this generally does not solve the problem of 13C-13C couplings." This sentence is inaccurate: standard 2-ketobutyrate, 2-ketoisovalerate, and acetolactate are all commercially available and used in standard applications, in forms where only one carbon is specifically labeled, which solves the 13C-13C coupling problem.**

   The referee is right. We rewrote the introduction to take this comment into account.

5. **Line 69: please indicate that 2-ketobutyrate is used for isotopic labeling of Ile-d1 methyl groups**

We have amended the introduction accordingly.

6. **Line 247: please state why perdeuterated pyruvate was used in combination with acetolactate to produce a sample specifically labeled on the Pro-S methyl groups.**

The mention of pyruvate in the queried sentence was incorrect. As described in the Experimental section, in this experiment deuterated pyruvate was used as the carbon source to grow the perdeuterated cells that were subsequently processed into eCells. Pyruvate was not used during eCell CFPS. We thank the referee for the query.

7. **Figures 3 and 4: please add examples of 1D traces at appropriate locations to show whether or not there is residual labeling for valine, leucine or pro-R methyl groups.**

There is evidence for undesired methyl cross-peaks at 10- to 30-fold lower contour levels. We now supply 2D spectra plotted at lower contour levels in the Supporting Information.

Referee 3:

1. **The described eCell system is obviously based on bacterial protoplasts stabilized by various layers of biopolymers. Therefore, the cell environment is left intact and the eCells should then not be called a cell-free system. Furthermore, I am mostly concerned about the various comparisons with real cell-free expression systems, as arguments sometimes appear to be biased and based on wrong facts. I would appreciate if a more objective discussion could be presented. Generally, the labeling approach was exemplified with two small, easy to synthesize and stable proteins. These targets do not have problematic signal overlap. It would thus be more convincing, if a more challenging target would have been included as proof of principle as well.**

The eCell technology has been published previously as a cell-free method (Van Raad and Huber, 2021). Although eCells are derived from live cells, they are in fact capsules that can neither replicate nor retain low-molecular-weight compounds, in close analogy to cell-free protein synthesis (CFPS) conducted in dialysis systems. We feel that the term 'eCell CFPS' adequately reflects the cellular nature of eCells.

In the updated version of the manuscript, we took pain not to compare the performance of eCell CFPS with established cell-free expression systems, as there are many different versions of CFPS. We performed direct experimental comparisons previously (Van Raad and Huber, 2021), but not in the present work.

The present work was conducted to establish proof-of-principle that eCells maintain the enzymatic activities required for amino acid biosynthesis. This is now mentioned in the revised introduction on line 76. We note that some methyl cross-peaks overlap even in the 13C-HSQC spectrum of ubiquitin and overlap would be significant in PpiB if all ILV methyls were detected in the same spectrum.

2. **Page 1/Abstract: The system…is more readily scalable than .. cell-free systems. I do not see any reason for this statement. CF system in the simple batch configuration can be scaled up even to multiple litre volumes (Zawada et al., 2022 Curr Opin Biotechnol). The two compartment configuration can be scaled from few µL reactions up to > 10 mL reactions.**

eCells can be prepared quickly and without the need of specialized equipment required for the preparation of cell lysates. For example, there is no need for mechanically breaking bacterial cells (by means of a homogenizer, French press or similar equipment), and high-speed centrifugation and dialysis steps as in S30 extract preparations are not necessary. As a result, the eCell system can readily be scaled up in volume. We attempted to make this clearer in the revised Discussion section (line 287). In addition, we simplified the statement in the Abstract to 'is readily scalable'.

3. **Furthermore, frequent targets where L/I/V signal overlap will really become problematic are membrane proteins. How will those be synthesized in eCells? As supplied hydrophobic environments cannot support their folding in the cell lumen as it is an option in CF systems, they must become translocated into the membrane. Therefore, same difficulties will most likely appear as known with membrane protein expression in intact *E. coli* cells.**

We have no experience with membrane proteins and haven't tested the eCell system with membrane proteins. Membrane proteins are beyond the scope of this study.

4. **Page 2, Line 51 ff: Truncation of amino acid biosynthetic pathways and reduced complexity in CF systems is often a benefit for labeling, not a disadvantage. E.g., Val and Leu can be individually labeled by providing KIV or MOV as precursors. In the eCell system, providing KIV would result in the labeling of both amino acids. Furthermore, all metabolic scrambling reactions of labeled amino acids still exit in the eCell system, whereas they are reduced in the less complex CF lysates.**

We agree with the reviewer that, depending on the application, non-functional biosynthetic pathways can be a benefit. We conducted the present work to show that eCells maintain the activity of enzymes involved in amino acid biosynthesis, thus allowing the use of simple precursors. Metabolic scrambling does occur but, as shown in the present manuscript, is limited for the biosynthetic pathways to valine and leucine (see Figures S4–S6).

5. **Page 15, Line 280 ff: eCells are compatible with toxic proteins. This is unlikely and has to be shown. E.g. any toxin that affects the cell membrane will most likely kill the system.**

We do not claim that every toxic protein can be produced with eCells. To account for this, we amended the text by changing 'the compatibility with toxic proteins' to 'compatibility with toxic proteins' (line 291). Notably, we have successfully used the eCell system to produce protein containing the anti-bacterial amino acid 3-fluoro-L-alanine,

which inhibits alanine racemase and thus interferes with cell-wall maintenance (https://doi.org/10.1021/ja00434a030). We did not include this result in the present manuscript, as a thorough exploration of the compatibility of different toxic proteins with the eCell system goes beyond the scope of the present work.

6. **..facile modulation of solution conditions for protein production. What is meant with that? In contrast to CF systems, the reaction compartment, i.e. the cell lumen of eCells, is not accessible for anything that cannot pass the membrane. CF reactions can be performed in presence of supplied chaperones, ligands, detergents, nanodiscs… I do not see any possibility for this with eCells.**

   The reviewer is correct that the interior of eCells is not readily accessible to macromolecules, but eCell conditions can easily be adjusted for different redox potentials, pH, cofactors etc. The simplest way of adding macromolecules to the interior of eCells would be their production during bacterial cell growth prior to encapsulation. We agree that nanodiscs may be incompatible with the eCell system. In the revised manuscript, we now refer to 'facile modification of the solution conditions with regard to compounds small enough to enter the eCells' (line 292).

7. **Importantly, eCells can be produced much faster than .. cell extracts…. Depending on the volume, CF lysate production takes one day of fermentation and processing, and another day for dialysis. In addition, much faster procedures have been published (Krinsky et al., 2016, Plos One). CF lysate aliquots are stable at minus 80 degrees for years and can be used for the synthesis of ANY protein target. With eCells, for each new target, a new eCell batch has to be prepared. Furthermore lifetime of eCells seems to be confined to few hours. This makes expression screening very tedious.**

   In the revised version, we changed the sentence to 'Importantly, eCells can be produced rapidly and easily.' (line 292). We explain why in line 287: 'In contrast to the preparation of cell extracts by mechanical lysis and high-speed centrifugation'. The method by Krinsky et al. (PLOS One, 11, e0165137, https://doi.org/10.1371/journal.pone.0165137, 2016) still requires mechanical cell lysis and high-speed centrifugation, and the protein yields reported were modest.

   eCells remain fully functional for years when stored at -80°C. This is now stated in line 293. Their lifetime is limited to several hours only under the conditions of cell-free protein synthesis at 37 $^O$C, similar to conventional bacterial cell lysates.

   Screening of different DNA constructs is not the topic of the present manuscript.

8. **Page 18, Line 352: ..the open nature of eCells… It is not clear up to which molecular mass substance can penetrate eCells. I assume that only works for LMW compounds? In that case, they do not have an open nature. Please comment and specify.**

   eCells leak GFP with a half-life time of about 4 hours (Van Raad and Huber, 2021). We have no quantitative data regarding the exchange rates of small molecules, but it is clear from the CFPS reactions that LMW compounds traverse into eCells easily. In response to

the referee's comment, we modified the sentence in the Conclusion section to 'The protocol for eCell preparation is uncomplicated, and the ready accessibility of the interior space of eCells to low-molecular-weight compounds provides ready control over the chemical environment, so that different isotope labelling is achieved simply by use of different reaction buffers.'

9. **Page 6, line 131: Is there a reason to use PEG-8000? I understand that PEG-8000 will not enter eCells and thus deplete the cells of water (or $D_2O$) in this case?**

We used PEG-8000 as PEG is a component of the Cytomim system by Jewett and Swartz (2004) but haven't explored its benefit in eCell CFPS and don't know whether it enters the eCells. We speculate that PEG-8000 helps prevent osmotic rupture of eCells, but this has not been tested.

10. **Page 7, line 154: Ubiquitin was produced from 300 mg eCells. Are the stated 300 mg or 800 mg eCells the dry pellet weight or is this eCells with alginate? What volume of eCells in buffer is provided? Is this the whole amount that is produced from 450 mL M9 D2O medium? If not, what is the amount of eCells that is obtained from one cultivation?**

Following encapsulation, eCells readily sediment with mild or even without centrifugation. The weight measurements were taken of the encapsulated cells prior to freezing. Lysis occurs upon thawing. We have now made this more explicit in lines 155–156.

11. **Page 9, line 200: The protein yield was 0.7 mg from 10 mL eCell CFPS reaction. For such easy proteins the yield is rather low. CF systems are reported to synthesize > 5 mg of protein in 1 mL. Wouldn't it be reasonable to use more than 300 mg eCells or to conduct a larger scale expression? The usual expression of ubiquitin in *E. coli* in even deuterated M9 is up to 50 mg/L. Thus, 0.5 L medium would amount around 25 mg protein. What is the total ubiquitin amount that can be obtained from eCells produced in 500 mL medium?**

300 mg eCells generated 3.8 mg of ubiquitin. Typically, 1 g eCell capsules can be obtained from 0.5 L cell culture, corresponding to about 12 mg ubiquitin produced from 1 g eCells. In a previous article (Van Raad and Huber, 2021), we compared eCells with our standard in-house continuous exchange CFPS method for the protein PpiB. The eCells yielded 45% less PpiB than the standard CFPS method.
We see no advantage in using larger scale expressions than needed to record NMR data. At present, we have little experience with the expression yields of other proteins.

12. **Page 16/Table 1: Table 1 ignores that cells were grown in deuterated medium, which adds to the costs. Peak doubling will increasingly become problematic when working with proteins of higher complexity. Furthermore. Yields of 3.8 mg with a labeling efficiency of 10% means 0.38 mg/mL of relevant protein. In addition, acetolactate labeling is higher than 9 $ per sample, if you add 3.5 mM deuterated single amino acids to the mix ? It would be more helpful to show total production costs of labeled protein,**

**including deuteration, total protein yield, labeling efficiency and also the use of single deuterated amino acids or NTPs. Alternatively, the field "Cost per reaction" should be changed to "Cost of precursor for one reaction". yield of labeled protein only, instead of showing yield of unlabeled ubiquitin.**

In the revised version, Table S1 explicitly lists the amino acids for the perdeuteration experiment and their costs. We also updated Table 1 as requested and report the cost of $D_2O$ required for *in vivo* growth of the *E. coli* cells (lines 330 and 345).

As stated in the revised manuscript (lines 82 and 301), the objective of the experiment conducted with biosynthetic fractional $^{13}C$-labelling was to explore whether eCells maintain the biosynthetic pathway required to produce valine and leucine from glucose under CFPS conditions. We leave it to the reader to decide whether the experiment is worthwhile in a specific situation.

Thank you again for considering our manuscript for publication in *Magnetic Resonance*.

---

## Author Response (AR2)

We thank reviewer 2 and the editor for the additional comments. In response to the specific points raised, we have prepared a revised version of the manuscript as outlined below.

From Reviewer 2:
1. p17, line 336: Amino acids with stereospecific 13 C enrichment of single methyl groups are related to SAIL amino acids. They are expensive but commercially available. They are produced by a former spin-off of the Kainosho laboratory and distributed by Cortec Net.

The statement has been revised to address the reviewer's comment regarding the availability of the amino acids.

2. Table S1. The price indicated for L-serine is for 200 mg and not for 1 gr. For several amino acids (Val, Asp, Glu, Phe, ..) the price corresponds to a racemic DL mixture. Are D-amino acids converted and incorporated into proteins? If not authors should specify whether they recommend doubling the amounts required compared to the use of L-amino acids. Alternatively, authors can provide the cost of L-amino acids in the table S1. Based on the changes, the authors should update the deuteration cost estimate in the manuscript.

The amino acid list has been amended in accordance with the reviewer's request to address concerns regarding pricing and the presence of racemic DL mixtures. Furthermore, the manuscript's deuteration cost estimate has been updated based on these revisions.

Additional suggestions from editor:
1. P. 6, L112. Provide force of centrifugation in units of g.
The force of centrifugation has been provided in units of g.

2. P. 5, L113 and 115. Provide weight of cells and volume for resuspension.
The weight of cells and the volume for resuspension have been included.

3. P. 7 L155. Provide weight of cells and volume for resuspension.
The weight of cells and the volume for resuspension have been specified.

4. P. 8. L208. Either remove the word high, or provide information of what it is high in relation to, with appropriate statistical test.
The word "high" has been removed

5. (P11 L231) To improve clarity use eCell CFPS instead of CFPS when referring to your method and only CFPS when referring to use of cell extracts. Revise throughout. For example the sentence on p. 11 L. 244-247, it is unclear if this was done here using eCell CFPS or previously using CFPS by the cited work.
This clarification has been applied throughout the manuscript.

6. P. 14 L275. Note the amount of glucose needed or remove the statement.
The amount of glucose needed has been noted

7. P. 15 L290. Change to "some of the advantages" as it does not inherit all advantages, as also noted by the reviewers.
The phrase has been modified to "some of the advantages" to acknowledge that not all advantages are inherited, as noted by the reviewers.

8. P. 15 L291. Either remove "compatibility with toxic proteins" or provide a reference for eCells. Alternatively change wording to clarify that this is a speculation, i.e. "likely to also be compatible with toxic proteins".

The wording regarding "compatibility with toxic proteins" has been revised to clarify that it is speculative

**9. P. 15 L293. Please add "likely" or similar or provide reference or data to support the statement.**
The word "likely" has been added

**10. P16 L305. Clarify "extraordinarily low cost" by providing what this is relative to and justify relevance of comparison as appropriate. It is confusing that it is extraordinary, and also inefficient and in fact alternative methods are preferred by the authors.**
The phrase "extraordinarily low cost" has been clarified.

**11. P. 18 L. 375. Please provide data for and clarify "too long either".**
"too long either" has been clarified.

**12. P. 19. L. 383. Remove second "ready".**
The second occurrence of the word "ready" has been removed.

Finally, I agree with reviewer-1's request that:
"Figures 3 and 4: please add examples of 1D traces at appropriate locations to show whether or not there is residual 2abelling for valine, leucine or pro-R methyl groups."

I appreciate that lower contours are provided, but 1D traces provide a more clear visual understanding of the S/N and are simpler for visual comparison of heights than contour plots. This can be provided in the SI section.

The requested examples of 1D traces demonstrating residual labeling for valine, leucine, or pro-R methyl groups have been included in the supplementary data.